# Detection of *Sargassum* from Sentinel Satellite Sensors Using Deep Learning Approach

Marine Laval [1,2,*], Abdelbadie Belmouhcine [3], Luc Courtrai [3], Jacques Descloitres [4], Adán Salazar-Garibay [5], Léa Schamberger [6,7], Audrey Minghelli [6,7], Thierry Thibaut [2], René Dorville [1], Camille Mazoyer [2], Pascal Zongo [1] and Cristèle Chevalier [2]

[1] Laboratoire des Matériaux et Molécules en Milieu Agressif (L3MA), Université des Antilles, 97275 Schoelcher, France
[2] Mediterranean Institute of Oceanography (MIO), IRD, Aix Marseille Université, CNRS, Université de Toulon, 13288 Marseille, France
[3] IRISA, Université de Bretagne Sud, 56000 Vannes, France
[4] AERIS/ICARE Data and Services Center, University of Lille, CNRS, CNES, UMS 2877, 59000 Lille, France
[5] Mexican Space Agency (AEM), Ciudad de Mexico 01020, Mexico
[6] Laboratoire d'Informatique et Système (LIS), Université de Toulon, CNRS UMR 7020, 83041 Toulon, France
[7] Laboratoire d'Informatique et Système (LIS), Aix Marseille Université, 13288 Marseille, France
* Correspondence: marine.laval@mio.osupytheas.fr

**Abstract:** Since 2011, the proliferation of brown macro-algae of the genus *Sargassum* has considerably increased in the North Tropical Atlantic Sea, all the way from the Gulf of Guinea to the Caribbean Sea and the Gulf of Mexico. The large amount of *Sargassum* aggregations in that area cause major beaching events, which have a significant impact on the local economy and the environment and are starting to present a real threat to public health. In such a context, it is crucial to collect spatial and temporal data of *Sargassum* aggregations to understand their dynamics and predict stranding. Lately, indexes based on satellite imagery such as the Maximum Chlorophyll Index (MCI) or the Alternative Floating Algae Index (AFAI), have been developed and used to detect these *Sargassum* aggregations. However, their accuracy is questionable as they tend to detect various non-*Sargassum* features. To overcome false positive detection biases encountered by the index-thresholding methods, we developed two new deep learning models specific for *Sargassum* detection based on an encoder–decoder convolutional neural network (CNN). One was tuned to spectral bands from the multispectral instrument (MSI) onboard Sentinel-2 satellites and the other to the Ocean and Land Colour Instrument (OLCI) onboard Sentinel-3 satellites. This specific new approach outperformed previous generalist deep learning models, such as ErisNet, UNet, and SegNet, in the detection of *Sargassum* from satellite images with the same training, with an F1-score of 0.88 using MSI images, and 0.76 using OLCI images. Indeed, the proposed CNN considered neighbor pixels, unlike ErisNet, and had fewer reduction levels than UNet and SegNet, allowing filiform objects such as *Sargassum* aggregations to be detected. Using both spectral and spatial features, it also yielded a better detection performance compared to algal index-based techniques. The CNN method proposed here recognizes new small aggregations that were previously undetected, provides more complete structures, and has a lower false-positive detection rate.

**Keywords:** ocean color; *Sargassum*; MODIS; MSI; OLCI; Sentinel-2; Sentinel-3; convolutional neural network; deep learning



## 1. Introduction

The brown macro-algae, *Sargassum,* from the genus *Sargassum* C. Agardh (Phaeophyceae, Fucales) is a type of algae commonly found in the Sargasso Sea (North Atlantic Ocean). However, since 2011, these *Sargassum* have started proliferating outside the Sargasso Sea, expanding to the Gulf of Guinea, the Caribbean Sea, and the Gulf of Mexico,

forming the Great Atlantic *Sargassum* Belt [1]. Three morphotypes of the same taxon are currently recognized within the Great Atlantic *Sargassum* Belt [2–4]: *Sargassum natans* I Parr, *S. natans* VIII Parra, and *S. fluitans* III Parr. *Sargassum* aggregate and form windrows or mats up to a length of hundred meters [5]. The significant amount of *Sargassum* aggregations in the area causes sizeable beaching events on the coast [6,7], consequently impacting the local economy [8,9], the environment [10–12] and causing major sanitary issues due to the emanation of harmful gas during the decomposition [13–15].

In order to predict and thus anticipate beaching events, it is crucial to collect spatio-temporal data about *Sargassum* aggregations. Several algal indexes were developed to detect *Sargassum* from satellite imagery, such as the Maximum Chlorophyll Index (MCI) [16] or the Floating Algae Index (FAI) [17]. Recently, the FAI was improved by the Alternative Floating Algae Index, AFAI [18], which is less sensitive to clouds. Using the MCI, Gower et al. [19] initiated the detection of *Sargassum* aggregations in the Gulf of Mexico using satellite images from the medium-resolution imaging spectrometer (MERIS) on board ESA's Envisat satellite and the moderate-resolution imaging spectro-radiometer (MODIS) on board NASA's Terra and Aqua satellites. It was then extended to the Ocean and Land Colour Instrument (OLCI) [20] on board Copernicus's Sentinel-3 satellites. The AFAI proved to be performant on MODIS [18] or the multispectral instrument (MSI) [5,7,21,22] on board Copernicus's Sentinel-2 satellites. These indexes can also be used to determine the coverage of *Sargassum* per pixel [18] and then the biomass quantity of *Sargassum* as empirically defined by Wang and Hu [23].

However, the standard index-based detection (ID) method, from for example MCI or AFAI, cannot discards all radiometric noise and other non-*Sargassum* factors (i.e., residual clouds and cloud shadow, land, coastal water, surface waves, and sunglint [18,24]), which interfere with the detection of low-coverage *Sargassum*. The standard method is based on the calculation of the background index and the use of a threshold on the difference between local and background indexes to determine whether a pixel contains Sargassum (above the threshold) or not (under the threshold). However, the background calculation itself relies on the accurate screening of clouds, cloud shadows, coastal areas, land and sunglint [18,21] that may yield false positive values. Finally, uncertainties in the calculation of the background index may induce non-significant positive values. As a result, the lowest index values are not significant enough to ensure the presence of *Sargassum* and must be discarded, inevitably yielding false negatives. In order to retain most *Sargassum* while rejecting most false detections, the threshold applied on indexes must be optimally determined [18,19]. While some false detections can be removed by mathematical morphology post-treatments (e.g., erosion-dilation [18,25–28]) false detections remain, linked to residual clouds, cloud shadows, sunglint and turbidity. Another limitation encountered by the ID method is the high cloud cover in some images, because in such cases the background index, hence the index, cannot be accurately estimated. Wang and Hu [18,21] estimated false positive detections for their retrieval to be 20% of the total *Sargassum* pixels for MODIS and 6% for MSI. Recently, a machine learning model revealed a higher rate, namely 50%, of false positive detections by the ID method applied to MODIS images [24]. The remaining false positive detections cannot be discriminated on their sole index value, since their index distribution overlaps with that of true *Sargassum*. The ID method also produces false negative detections (i.e., *Sargassum* pixels not detected); it has been estimated to be 7% for MODIS [18] and 20% for MSI [21]. Therefore, the ID method has two important limitations: the need for complex pre-processing and the unavoidable yield of false detections (false positive and false negative).

An alternative to the ID method can be neural networks. Indeed, in the last few years neural networks have made significant progress in computer vision and are widely used in several domains, including imagery in oceanology [29–33]. Furthermore, they have already been successfully used for other macro-algae detections as an alternative to the ID method [34–36]. *Sargassum* in near-shore water and *Sargassum* stranding on the Atlantic coast have also been investigated using classic classifier algorithms (random forest,

K-means) and deep neural networks (UNet, AlexNet, VGG . . . ) on the Mexican-Caribbean and Caribbean coasts using smartphone camera pictures [37,38] and fixed cameras [39]. On coastal and off-shore waters, machine learning has also been used for *Sargassum* detection from satellite images, e.g., random forest for false detection extractions [24] or a combination of several indexes [40] to improve *Sargassum* detection in the Atlantic Ocean. In the Yellow Sea, next to China and Korea, *Sargassum* detection was also studied with several machine learning models (i.e., fine tree, support vector machine, etc.) [41,42].

Among neural network methods, deep neural networks, such as convolutional neural networks (CNN), are efficient for image processing in object segmentation and classification [43–47]. Contrary to the previous machine learning algorithms, these types of neural networks could improve open sea *Sargassum* detection in the Atlantic Ocean by automatically learning the relationship between the reflectance and the spatial context of *Sargassum* aggregations. Arellano-Verdejo et al. [48] proposed a first deep learning method for *Sargassum* detection on the Mexican Caribbean coast, ErisNet, i.e., a pixel classifier with convolutional and recurrent layers, applied to a selection of MODIS spectral bands. ErisNet weakly takes into account neighbor pixels, thus it does not learn the structure of the *Sargassum* aggregations composed of several pixels. Recently, Wang and Hu [49] used the UNet model [47] using FAI and RGB images from high resolution satellite data such as MSI, Operational Land Imager, WorldView-II, and PlanetScope/Dove to segment *Sargassum* in off-shore waters of the Lesser Antilles and the Gulf of Mexico. UNet is a CNN using an encoder-decoder style network and this architecture allows *Sargassum* aggregation structures to be learnt. However, the spectral information can be reduced by using only three spectral bands when the information provided by other available spectral bands might be used.

To avoid all those problems and increase the performance of detection compared to previous methods, we propose two new specialized models for *Sargassum* detection based on deep neural learning in the Caribbean and Atlantic areas. Those models used all the available spectral bands provided by a sensor contrary to ID methods and UNet [49]. Furthermore, deep neural networks, and, more specifically, CNN, have a higher abstraction level than simple machine learning algorithms, and are better structured to take into account neighbor pixels at a higher scale. Thus, the CNN models are based on an encoder–decoder architecture, contrary to ErisNet. MSI and OLCI sensors were chosen due to the difference of their spatial and spectral resolutions, but CNNs can be applied to other similar sensors.

This paper is organized as follows: first, the satellite data, the study area, the previous detection methods and our CNN architecture, training and validation methods are presented in Section 2. Second, the performance of our CNN models is quantitatively analyzed in Section 3. Third, the *Sargassum* detection obtained using our CNN models and ID methods are compared from the literature images in Section 4. Finally, a conclusion is proposed in Section 5.

## 2. Materials and Methods

### 2.1. Satellite Data

This study used images from several satellite sensors with different spatial resolutions to detect *Sargassum* from space: MSI onboard Sentinel-2, OLCI onboard Sentinel-3 and MODIS onboard Terra satellites.

The Sentinel-2 and Sentinel-3 missions are carried out by the European Union's Copernicus Earth Observation programme, and each mission is currently composed of a constellation of two satellites.

The Sentinel-2A and Sentinel-2B satellites were launched in 2015 and 2017, respectively. These satellites carry the multi-spectral instrument (MSI), which acquires data in 13 multispectral bands in the visible, near-infrared, and short-wave infrared ranges of the solar spectrum (0.44 μm to 2.19 μm), with a high spatial resolution (10 m, 20 m, and 60 m, depending on the bands). In this study, images have a resolution of 20 m. Each MSI sensor has a revisit period of five days.

Onboard the two Sentinel-3 satellites (Sentinel-3A and Sentinel-3B, launched in 2016 and 2018, respectively), the Ocean Land Color Instrument (OLCI) has similar spectral bands than the MERIS sensor (2002–2012). OLCI images are composed of 21 spectral bands from the visible to the near infrared (from 0.4 μm to 1.02 μm), with a spatial resolution of 300 m. Each OLCI sensor has a revisit period of two days.

The Moderate Resolution Imaging Spectroradiometer (MODIS) instrument flies on board the Terra and Aqua satellites launched by NASA in 1999 and 2002 respectively. MODIS images have 36 spectral bands (0.4 μm to 14.4 μm) at three spatial resolutions (250 m, 500 m, and 1 km). Each MODIS sensor has a revisit period of one to two days. Only the MODIS/Terra sensor was used in this study because the overpass time is closer to the Sentinel satellites than MODIS/Aqua.

### 2.2. Study Area and Data Set

The west coast of the Atlantic is particularly prone to massive arrivals of *Sargassum* aggregations from the Atlantic Ocean since 2011 [1,50]. The study area is located in the Lesser Antilles, in the East Caribbean Sea and the Mexican coast (Figure 1), from 2018 to 2022.

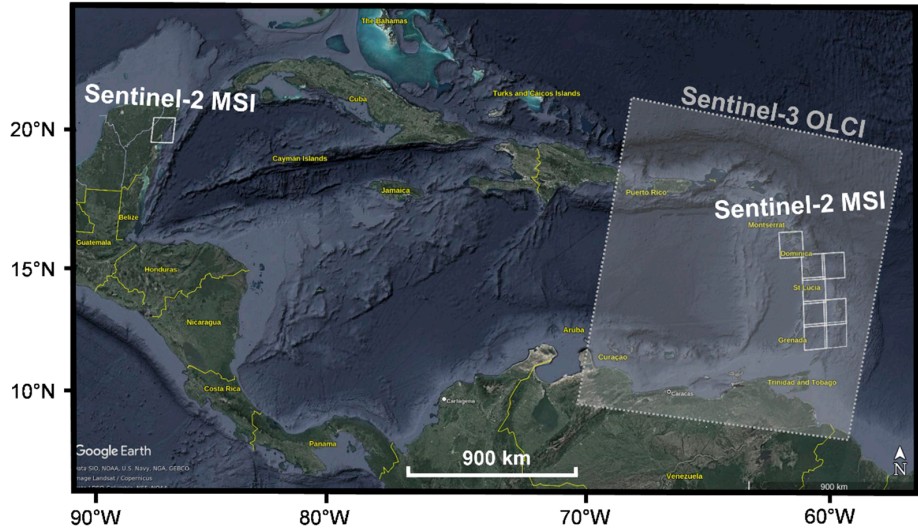

**Figure 1.** Location of Sentinel-3 OLCI images (white dotted boxes) and Sentinel-2 MSI tiles (white boxes) used for the learning and focus sets.

This study focused on two products: level-2A MSI products and Level-2 OLCI products from Copernicus Open Access Hub [51]. All images were corrected for Rayleigh scattering, gaseous absorption, and aerosols and have their corresponding ground truth. Images of each spectral band were normalized using their mean and standard deviations. Then, those images were divided into two types of datasets: a learning dataset and a focus dataset.

The learning dataset targeted CNN learning (training, validation, and test) and the quantitative validation process for our proposed CNNs. This dataset was composed of 14 images of size 10,980 × 10,980 pixels from MSI (Table S1) and 16 images of size 4865 × 4091 pixels from OLCI images (Table S2) from 2017 to 2022 distributed across all the study areas (Lesser Antilles and the Mexican Caribbean coast). All spectral bands were used as inputs to the model. To increase the learning dataset and avoid overfitting, we generated more images by randomly applying a horizontal flip, a vertical flip, or a rotation of 90 or 270 degrees. Additionally, a random color augmentation was applied only to OLCI images because they have more significant pixel variations than MSI images, due to the variation in the sunlight intensity from east to west. Indeed, at most, three spectral bands were picked randomly and adjusted by random values selected from [−0.1,0.1]. *Sargassum*

represented, on average, only 1% of the pixels of an image. Thus, 128 × 128 sub-images centered on random positions were taken while conserving the proportion of *Sargassum* in the original images. We ensured that 66% of the sub-images had no *Sargassum* to provide a representative sample of different backgrounds. Therefore, the dataset comprised 75,000 sub-images from MSI and 50,000 sub-images from OLCI. The learning dataset was split into three subsets: a training set including 80% of the total image number, a validation set comprising 10% of the images, and a test set containing the remaining images. The test set was only used to evaluate the model's performance.

A focus dataset was then built in order to: (1) quantitatively evaluate the CNN models performance; and (2) discuss the performance compared to the ID method images published and validated in the literature. That dataset was composed of complete images, (not sub-images) from MSI and OLCI sensors, detailed in Table 1. It contained four images of level-1 files from the literature (Table 1): MODIS and MSI images from Descloitres et al. [22] and OLCI images from Schamberger et al. [52]. The data also included our CNN images from the OLCI sensor at the same dates and locations and from the MSI sensor only at the same dates and locations as those of Descloitres et al. [22]. In addition, the dataset contained a supplemental image from MSI on 6 August 2022 in the Martinique area.

**Table 1.** Images of the focus dataset used to evaluate the performance of our CNNs compared to images from the literature, with a summary of their characteristics: location, acquisition date and time, satellite, and sensor. The 'Source' column indicates the source of each image: a reference for images taken from the literature, and 'us' for images processed with our own CNN method.

| Location | Date | Time (UTC) | Satellite-Sensor | Source |
|---|---|---|---|---|
| Grenadines (Caribbean) | 29 January 2019 | 14:35 | Terra-MODIS | [22] |
| | | 14:37 | Sentinel-2B-MSI tiles: PQU- PQV-PRU-PRV | [22], us |
| | | 13:38 | Sentinel-3-OLCI | us |
| Lesser Antilles | 9 May 2020 | 13:56 | Sentinel-3-OLCI | [52], us |
| | 8 July 2017 | 13:55 | Sentinel-3-OLCI | |
| Martinique | 6 August 2022 | 14:37 | Sentinel-2-MSI tile PRB | us |

*2.3. Three Sargassum Detection Methods*

2.3.1. Standard Index-Thresholding Method

Currently, the satellites cannot detect the differences between the three morphotypes of *Sargassum* living in sympatry. Therefore, they are studied as a whole. Two indexes were essentially used in this study: the Maximum Chlorophyll Index (MCI) [16] and the Alternative Floating Algae Index (AFAI) [18]. Furthermore, the Floating Algae Index (FAI) [17], the Normalized Difference Vegetative Index (NDVI) [53], and the InfraRed Color (IRC) were also used for the ground truth computation (Section 2.3.3).

The NDVI takes advantage of the red edge of the vegetation spectral reflectance. It measures the normalized difference between near-infrared ($\lambda_2$) and red range ($\lambda_1$):

$$\text{NDVI} = \frac{\text{R}(\lambda_2) - \text{R}(\lambda_1)}{\text{R}(\lambda_2) + \text{R}(\lambda_1)} \tag{1}$$

This index is often used to study vegetation ecology [54–56]. Floating algae have also been investigated using NDVI [57,58]. MCI, FAI and AFAI indexes were calculated using Equation (2), based on wavelengths summarized in Table 2. The MCI is an index based on radiance around 705 nm, more accurately, the MERIS and OLCI spectral bands located at 709 nm. MCI reveals high chlorophyll concentrations; it was first used to detect phytoplankton bloom using the MERIS sensor [16]. That index also showed its efficiency on *Sargassum* observations in the Atlantic Ocean using MERIS, MODIS and OLCI sensors [19,20]. The

FAI was developed for floating algae in the open ocean and firstly tested with the MODIS instrument to detect the benthic macroalgae *Ulva prolifera* in the Yellow Sea and pelagic *Sargassum* in the North Atlantic Ocean [17]. Later, Wang and Hu [18] improved it with the AFAI, an index more efficient in the presence of clouds. It was developed for the MODIS sensor to detect *Sargassum* and extended to the MSI sensor [22].

$$Index = R(\lambda_2) - R(\lambda_1) - [R(\lambda_3) - R(\lambda_1)] \times \frac{\lambda_2 - \lambda_1}{\lambda_3 - \lambda_1} \tag{2}$$

**Table 2.** Wavelength parameters of Equations (1) and (2) for NDVI, MCI, FAI and AFAI indexes for OLCI, MSI and MODIS.

| Index | Sensor | $\lambda_1$ (nm) | $\lambda_2$ (nm) | $\lambda_3$ (nm) |
|---|---|---|---|---|
| NDVI | OLCI | 665 | 865 | - |
| | MSI | 665 | 833 | - |
| MCI | OLCI [1] | 681 | 709 | 754 |
| FAI | MODIS [2] | 645 | 859 | 1240 |
| | MSI [3] | 655 | 855 | 1609 |
| AFAI | MODIS [4] | 667 | 748 | 869 |
| | MSI [5] | 665 | 740 | 865 |

[1] [20]; [2] [17]; [3] [21]; [4] [18]; [5] [22].

While MCI, FAI and AFAI can be directly used for *Sargassum* detection, additional processing is required to improve the detection. Indeed, while Equation (2) guarantees that the *Index* increases with the presence of *Sargassum*, it also increases with the value of the *Index* for *Sargassum*-free surrounding water, the so-called background *Index*. This background *Index* depends on local non-*Sargassum* factors, such as water spectral reflectance, potential residual sunglint contamination, and the presence of aerosols. Then, a local *Index* deviation (*δIndex*), i.e., the difference between the background and the local *Index*, is used to discriminate between pixels with and without *Sargassum* thanks to a threshold. This latter is optimally determined [18,19], but also depends on the author's objective and specific datasets; for example, the threshold used on OLCI images by Schamberger et al. [52] is 0.0030, while the one used on MODIS images by Descloitres et al. [22] is 0.00017 [18].

Finally, the amount of *Sargassum* may be determined with the fractional coverage (FC), it is defined as the ratio between the surface area of *Sargassum* within one pixel and the total surface of that pixel [22]. The FC is deduced thanks to the linearity of Equation (2), which ensures that the local *δIndex* is proportional to the FC of *Sargassum* within a sensor pixel [18,22]:

$$\delta Index(FC) = K \times FC \tag{3}$$

In this study we used K = 0.0874 for MODIS, K = 0.0824 for MSI [22] and K = 0.0579 for OLCI [59].

The biomass quantity of *Sargassum* can be estimated using a linear relationship between the fractional coverage, the pixel area and a calibration constant defined empirically by Wang and Hu [23].

### 2.3.2. Convolutional Neural Network (CNN) for Sargassum Retrieval

State of the Art

In this study, we used three other CNNs to compare with our method, including two CNNs already used for *Sargassum* extraction in the Atlantic and Caribbean areas since 2019.

These are the ErisNet model from Arellano-Verdejo et al. [48] and the UNet of Ronneberger et al. [47], adapted by Wang and Hu [49] for *Sargassum*, and by Yan et al. [60], for harmful cyanobacterial algal blooms detection.

ErisNet [48] contains nine convolutional layers and two recurrent layers, allowing the model to keep a history of previous predictions. The network learns using a balanced number of *Sargassum* and background pixels during the training. However, during inference, it takes all pixels and predicts for each of them whether they belong to *Sargassum* or not. ErisNet is a pixel classifier that processes pixels as 1D input, unlike UNet and SegNet, which are image segmenters and treat the image as 2D input.

The encoder–decoder style network UNet [47] was used by Wang and Hu [49] for *Sargassum* retrieval. It was initialized with the weights of a VGG16 [61] trained on ImageNet. UNet contains four convolutional layers and performs four down-samplings in the encoder and four up-samplings in the decoder. The UNet encoder groups pixels of the same types together, whereas the decoder magnifies the output of the encoder. Each decoded map is concatenated with the same-sized encoded map. UNet was also applied by Yan et al. [60] on sentinel-2 MSI images of Chaohu Lake in China to identify harmful cyanobacterial algal blooms (CyanoHABs). They compared the segmentation with three FAI-based automatic CyanoHABs extraction methods: gradient mode, fixed threshold, and the Otsu method, and showed that the accuracy of UNet was better.

SegNet [43] is another encoder-decoder-style segmentation network powerful for image segmentation. Like UNet, SegNet has reduction layers connected to enlargement layers to reconstruct pixels. In addition, it uses neighboring pixels, having the same distribution during the reduction, in the upsampling reconstruction. UNet and SegNet can process small images (here from $128 \times 128$) in a single pass.

We built these three models using descriptions found in the literature (parameters summarized in Table 3), and trained them with the same dataset as our own models.

**Table 3.** Summary of the layer, block and total parameter numbers for ErisNet, UNet, SegNet and our proposed Networks for MSI and OLCI. More detailed tables of the structure of our proposed Networks are available in Supplementary Materials for MSI (Table S3) and OLCI (Table S4).

| | ErisNet | UNet | SegNet | Our Proposed Network | |
| --- | --- | --- | --- | --- | --- |
| | | | | MSI | OLCI |
| Layers | 44 | 83 | 91 | 32 | 75 |
| Blocks | 7 | 10 | 10 | 9 | 18 |
| Total Parameters | 455,554 | 13,400,578 | 29,449,350 | 226,762 | 973,145 |

Model Description for MSI and OLCI

We proposed two convolutional neural networks (CNN) with an encoder–decoder architecture (parameters summarized in Table 3), one targeting *Sargassum* semantic segmentation and taking Sentinel-2/MSI images as input, while the other focused on Sentinel-3/OLCI images and returning the δMCI (post-processed without clouds, land, and coasts). Indeed, with the ID method, OLCI images have more diffuse *Sargassum* aggregations than MSI. Thus, it was more relevant to train the CNN on the δMCI than on semantic segmentation. For MSI and OLCI images, input channels were composed of all the image spectral bands listed in Tables S1 and S2. The code of the two models was written using Pytorch [62].

For the CNN with MSI images, the encoder reduces the image scale, and the decoder performs a semantic segmentation. Indeed, we made only a single reduction because *Sargassum* aggregations have a width of one to three pixels. Hence, too much reduction is useless since target objects are thin. The encoder is composed of residual blocks issued from ResNet [63]. Figure 2 shows the architecture of the network; each residual block conserves its input by adding it to its output. Since there is only a need to segment *Sargassum*, a weighted binary cross-entropy loss function (WBCEWithLogitsLoss) was used. This loss is appropriate for binary problems and helped to deal with the unbalanced problem of *Sargassum* because the ratio of *Sargassum* pixels to the background pixels was meager. Let

$y_{i,j}$ be a pixel of the ground truth segmentation, $\hat{y}_{i,j}$ be the output of the network, which corresponds to the logits (log-odds function) of the semantic segmentation, and $N \times M$ be the resolution. The weighted BCEWithLogitsLoss is defined by Equation (4):

$$\text{WBCEWithLogitsLoss}(y, \hat{y}) = -\frac{1}{N \times M} \sum_{i,j} \left[ w_1 \times y_{i,j} \times \log\left(\sigma\left(\hat{y}_{i,j}\right)\right) + w_2 \times \left(1 - y_{i,j} \times \log\left(1 - \sigma\left(\hat{y}_{i,j}\right)\right)\right) \right] \quad (4)$$

where $\sigma(x) = \frac{1}{1+e^{-x}}$ is the sigmoid function, $w_1$ is the weight of the foreground, $w_2$ is the weight of the background.

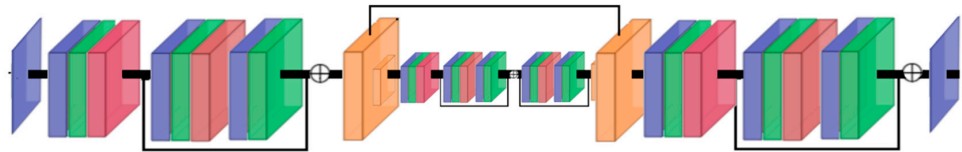

**Figure 2.** The neural network architecture for semantic segmentation on MSI; blue layers are convolutions, green ones represent batch normalizations, ReLU is in red, and the orange layers correspond to max pooling/unpooling layers.

Regarding OLCI, the segmentation map is substituted with the δMCI. The architecture of the network is similar to the one of MSI, except that it adds one reduction level and more channels in the residual blocks. The network contains 128, 256, and 512 blocks against 32 and 64 for MSI (Figure 3). Furthermore, a soft attention block (Attention UNet; [64]) is added to each reduction level. These blocks allow the connection between outputs of unpooling layers and inputs of reduction layers corresponding to the same scale. The goal is to use the distribution of the same scale-reduction input.

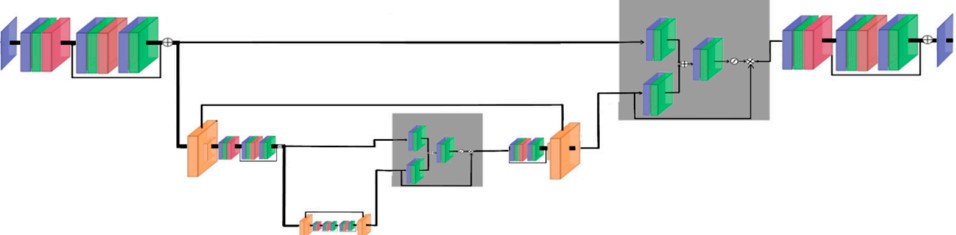

**Figure 3.** The neural network architecture for OLCI images; blue layers are convolutions, green ones represent batch normalizations, ReLU is in red, and yellow layers correspond to max pooling/unpooling layers. Gray blocks represent soft attention.

Because the task is a regression, the mean squared error (squared L2 norm) was used as a loss (MSELoss). This loss function compares the output δMCI image with the reference one. Let $\delta MCI_{ij}$ be a pixel of the reference masked δMCI, $\widehat{\delta MCI}_{i,j}$ be a pixel of the network's output filter, and $N \times M$ be the resolution. The MSELoss is written as follows:

$$\text{MSELoss}(\delta MCI, \widehat{\delta MCI}) = -\frac{1}{N \times M} \Sigma_{i,j} (\delta MCI_{i,j} - \widehat{\delta MCI}_{i,j})^2 \quad (5)$$

Training Process

For the training process, we used the training and the validation subsets from the learning dataset described in Section 2.2. The training lasted for 1000 epochs and was done using a batch size of 92 and a stochastic gradient descent (SGD) with a learning rate of $10^{-3}$ and a momentum of 0.9. Finally, the model giving the best F1-score on the validation set was saved. This validation was done every ten epochs.

### 2.3.3. Visual Analysis to Establish the Ground Truth

The ground truth was determined for each image of the learning set (i.e., training, validation and test datasets) and the focus dataset described in Section 2.2. In the absence of in situ observations of the *Sargassum* distribution, the ground truth was provided by an "expert" who manually annotated *Sargassum* aggregations using different standard indexes (i.e., FAI, AFAI, NDVI, MCI, IRC; Table 1) as indicators from level-2 OLCI and MSI images. These indexes were chosen due to their different response to *Sargassum* signals, and the sensor used.

The ground truth of the MSI was a binary mask: 1 for pixels containing *Sargassum* and 0 for *Sargassum*-free pixels. For OLCI, the ground truth for the training process (training and validation datasets) was obtained using a binary mask built manually using several indexes to identify *Sargassum*, then applied to the δMCI (Figure 4). For the model evaluation, we only compared OLCI detections to the binary ground truth (i.e., the accuracy of the retrieved δMCI values was not evaluated).

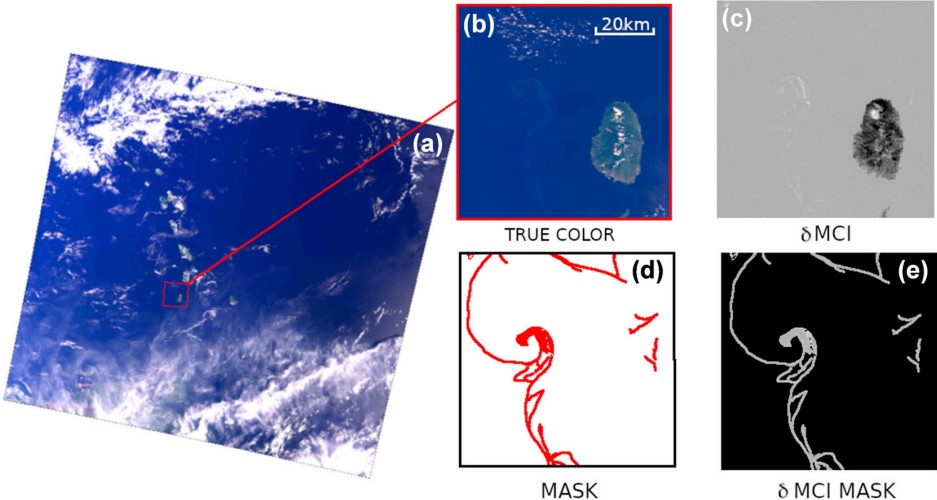

**Figure 4.** (**a**) OLCI image on 23 February 2017 with True color composite (band 9:674 nm:Red; band 7:560 nm:Green; Band 2:413 nm:Blue). (**b**) Sub-image example (red square); (**c**) δMCI; (**d**) binary mask manually made using several indexes and (**e**) masked δMCI built using the binary mask.

### 2.4. Performance Evaluation

### 2.4.1. Performance Metrics Using the Ground Truth

Three types of metrics were used to evaluate the performance of the ID or CNN methods by comparing the test dataset to the ground truth: the recall, the precision, and the F1-score. The latter was calculated from the two other metrics (the recall and the precision). These metrics depend on three parameters: true positive (TP), false positive (FP) and false negative (FN). TP, FP and FN were determined using the ground truth and *Sargassum* pixels detected by the retrieval technique. TPs are *Sargassum* pixels successfully recognized by the retrieval technique. FPs are *Sargassum* pixels only identified by the retrieval technique but not present in the ground truth. Finally, FNs are *Sargassum* pixels not detected by the retrieval method and present in the ground truth.

The recall metric is the ratio of the number of TP detections and the number of *Sargassum* pixels of the ground truth (Equation (6)). The precision metric is the ratio of the number of TP detections and the number of pixels detected as *Sargassum* (Equation (7)). To evaluate the model's performance, these two metrics were combined to form the F1-score metric (Equation (8)). The better the performance; the closer the F1-score is to 1. *Sargassum*

detection was considered a "true" detection when its distance from an annotation on the ground truth is below three pixels.

$$\text{Recall} = \frac{\text{TP}}{\text{TP} + \text{FN}} \tag{6}$$

$$\text{Precision} = \frac{\text{TP}}{\text{TP} + \text{FP}} \tag{7}$$

$$\text{F1\_score} = 2 \times \frac{\text{Recall} \ \times \ \text{Precision}}{\text{Recall} + \text{Precision}} \tag{8}$$

2.4.2. Comparison of CNN and ID Approaches

Another evaluation was performed on the focus dataset (described in Section 2.2) based on the comparison of CNN with ID method images in order to estimate their respective performance. During this analysis, we no longer took into account the ground truth, which was not without errors, due to a degree of subjectivity present in the process of annotating *Sargassum* aggregations, especially for aggregation edges and low-FC aggregations. Indeed, the ground truth can introduce biases during the model performance evaluation with the performance metrics (Section 2.4.1).

Before the evaluation, a threshold set to 0.01 was applied to the δMCI of the CNN on OLCI images in order to optimally discard δMCI not linked to *Sargassum* aggregations.

To compare *Sargassum* aggregations obtained using CNN and ID approaches, the aggregations were extracted with a polygon function from Matlab (bwlabel function). *Sargassum* aggregation features such as their area, major axis size (i.e., length), minor axis size (i.e., width) and major/minor axis ratio (i.e., length/width ratio) were then measured.

To study the accuracy of the aggregations extracted by the CNN, the main aggregations detected by the ID method were compared. The main ID-detected aggregations were selected to: (1) have a length/width ratio higher than three [49]; and (2) to be in the 90th percentile of the length distribution. Thus, these main aggregations were larger than 3000 m and 140 m length for OLCI and MSI, respectively. In addition, *Sargassum* aggregations near coasts were discarded because of the FC high values in such areas due to the turbidity and shallow water [7,24,65,66]. The distance was set to 15 km away from coasts for OLCI and 200 m for MSI images.

## 3. Results

### 3.1. Model Performances on the Test Dataset

The model performance metrics (precision, F1-score) were calculated. They were both relatively high for MSI and OLCI. *Sargassum* pixels were accurately detected, and there were few false positive detections. Consequently, the F1-score is significant for the two proposed CNNs (Table 4), with an F1-score higher for MSI (88% of *Sargassum* pixels accurately detected versus 76% for OLCI). The lower F1-score of OLCI can be explained by a lower precision (79% versus 94% for MSI). OLCI has sparser aggregations than the ground truth. As a result, it detects more *Sargassum* pixels and has a lower precision than MSI.

**Table 4.** Recall, precision and F1-score of each method (indexes and Neural Networks) for Sentinel-2/MSI and Sentinel-3/OLCI. For Neural Network these metrics are calculated from the test data set (from the learning set of Tables S1 and S2).

| | Sentinel-2/MSI | | | Sentinel-3/OLCI | | |
|---|---|---|---|---|---|---|
| | Recall | Precision | F1-Score | Recall | Precision | F1-Score |
| NDVI | 0.835 | 0.085 | 0.154 | 0.910 | 0.105 | 0.188 |
| FAI(MSI)/MCI(OLCI) | 0.587 | 0.120 | 0.200 | 0.619 | 0.167 | 0.320 |
| ErisNet | 0.958 | 0.179 | 0.302 | 0.896 | 0.314 | 0.465 |
| UNet | 0.618 | 0.818 | 0.704 | 0.961 | 0.452 | 0.615 |
| SegNet | 0.599 | 0.840 | 0.699 | 0.931 | 0.493 | 0.645 |
| Our network | 0.819 | 0.942 | 0.876 | 0.735 | 0.785 | 0.760 |

## 3.2. Comparison with the ID Method

The performance metrics were also calculated for the three indexes (NDVI and AFAI for MSI or MCI for OLCI; Table 4). The indexes are very sensitive to land contamination, which can bias their performance. The land was therefore masked out before computing the indexes. The recall was better than with the CNN, especially for the NDVI. This is not surprising because these indexes were used to create the ground truth. However, their precision was low, below 17%, leading to a low F1-score (lower than CNNs) for all of them. Indeed, regarding the ID method, many false detections occurred around clouds, and many isolated detections seemed not to be *Sargassum* aggregations.

## 3.3. Comparison with Existing Networks

For comparison purposes, the other networks (ErisNet, UNet, SegNet) were trained in the same conditions as the proposed CNN models. Note that the precision and the recall were computed using a zero threshold, and all non-zero outputs were considered positive.

Compared to other CNN models, ErisNet network had the lowest F1-score (0.30). *Sargassum* pixels were well detected (recall around 90% for OLCI and MSI). However, it had a substantial number of false positive detections (a precision of 18% and 31% for MSI and OLCI, respectively, (Table 4)) around cloud edges and in the open ocean; the network principle can explain this. ErisNet is not a segmentation network like the other, in the sense that it processes pixels independently as 1D inputs and learns one background pixel for one *Sargassum* pixel. It does not leverage information in all background pixels since *Sargassum* pixels represent just a tiny portion of all pixels. On the contrary, segmentation networks (UNet, SegNet, and ours) consider the neighboring pixels.

Among encoder–decoder networks, our proposed network had the highest F1-score with 0.88 versus ~0.70 for UNet and SegNet on MSI images, and 0.76 versus ~0.62 for OLCI images (Table 4). On MSI images, our proposed network had fewer false positive and false negative detections than SegNet and UNet, with a recall of 0.82 versus ~0.60, and a precision of 0.94 versus ~0.83. Regarding OLCI images, SegNet and UNet were more efficient at detecting *Sargassum* pixels (recall around ~0.94 for both SegNet and UNet, versus 0.74 for our proposed CNN). However, the proposed approach had significantly fewer false positive detections (precision of 0.79 versus ~0.47 for SegNet and UNet), which resulted in a higher F1-score. Indeed, the multiple levels of reduction in UNet and SegNet makes the detection of filiform objects such as *Sargassum* aggregations difficult.

## 3.4. Results with the Focus Image Dataset

The performance metrics were also computed for images of the focus dataset to evaluate their validity. For Sentinel-3, the two images tested had a F1-score of 0.72 and 0.79 (Table 5), i.e., a score around the value of the corresponding tested CNN (0.76; Table 4). For Sentinel-2, the F1-score was 0.90, 0.88, 0.87 and 0.81, respectively, for the PQV, PRU, and PRV tiles of 29 January 2019 and the PRB tiles of 6 August 2022. For those tiles, the F1-score was in the same range as the one found during the performance evaluation of the corresponding model (0.88; Table 4). Only the PQU tile was out of range, with an F1-score of 0.65. The results shown proved the consistency of our technique.

**Table 5.** Recall, Precision and F1-score from our models for some images of the focus dataset. Note that the results from 29 January 2019 images (MSI) are biased due to the use of their presence in the training set.

| Satellite-Sensor | Date | Tile | Recall | Precision | F1-Score |
|---|---|---|---|---|---|
| Sentinel-3-OLCI | 29 January 2019 | - | 0.566 | 0.970 | 0.715 |
| | 8 July 2017 | - | 0.851 | 0.739 | 0.791 |
| Sentinel-2B-MSI | 29 January 2019 | PQU | 0.495 | 0.951 | 0.651 |
| | | PQV | 0.833 | 0.980 | 0.900 |
| | | PRU | 0.786 | 0.999 | 0.880 |
| | | PRV | 0.832 | 0.904 | 0.867 |
| | 6 August 2022 | PRB | 0.817 | 0.896 | 0.854 |

## 4. Discussion

The focus dataset was used here to discuss the CNN images in relation to the ID method. First, the main aggregations were analyzed to verify the reliability of CNN. Thereafter, we compare all *Sargassum* pixels of the CNN with the ID method to explain their differences and evaluate the suspected false positive and false negative detections. Finally, we compared OLCI and MSI images from the CNN method.

### 4.1. The CNN Reliability on Sargassum Aggregations

The main *Sargassum* aggregations of the ID method were detected by our proposed CNN on OLCI images (Figure 5a) and MSI images (Figure 6a). The CNN detected more than 70% of the FC of main aggregations detected by the ID method (70% for OLCI and 80% for MSI). This corresponds, respectively, to 90% and 48% of *Sargassum* pixels belonging to the main aggregations of OLCI and MSI images. Note that the main aggregations are also composed of low FC. In that respect, the CNNs are robust for main aggregations.

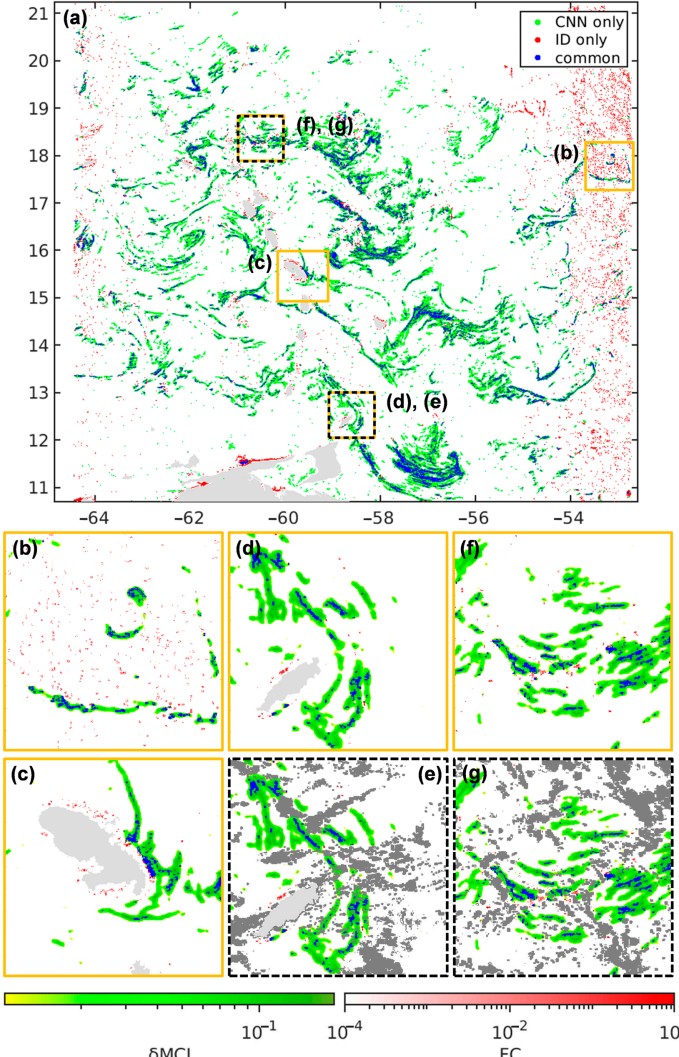

**Figure 5.** (**a**) *Sargassum* detections from OLCI with the CNN and ID methods: detected by both methods (blue), detected by the CNN but not by the ID method (green), and detected by the ID method but not by the CNN (red); (**b**–**g**) sub-images from (**a**) with a color scale for the ID and CNN detections; and (**e**,**g**) are respectively (**d**,**f**) with a cloud mask. MCI data for the calculation of the FC, cloud (dark gray) and land masks (light gray) come from Schamberger et al. [52]. OLCI image from 8 July 2017.

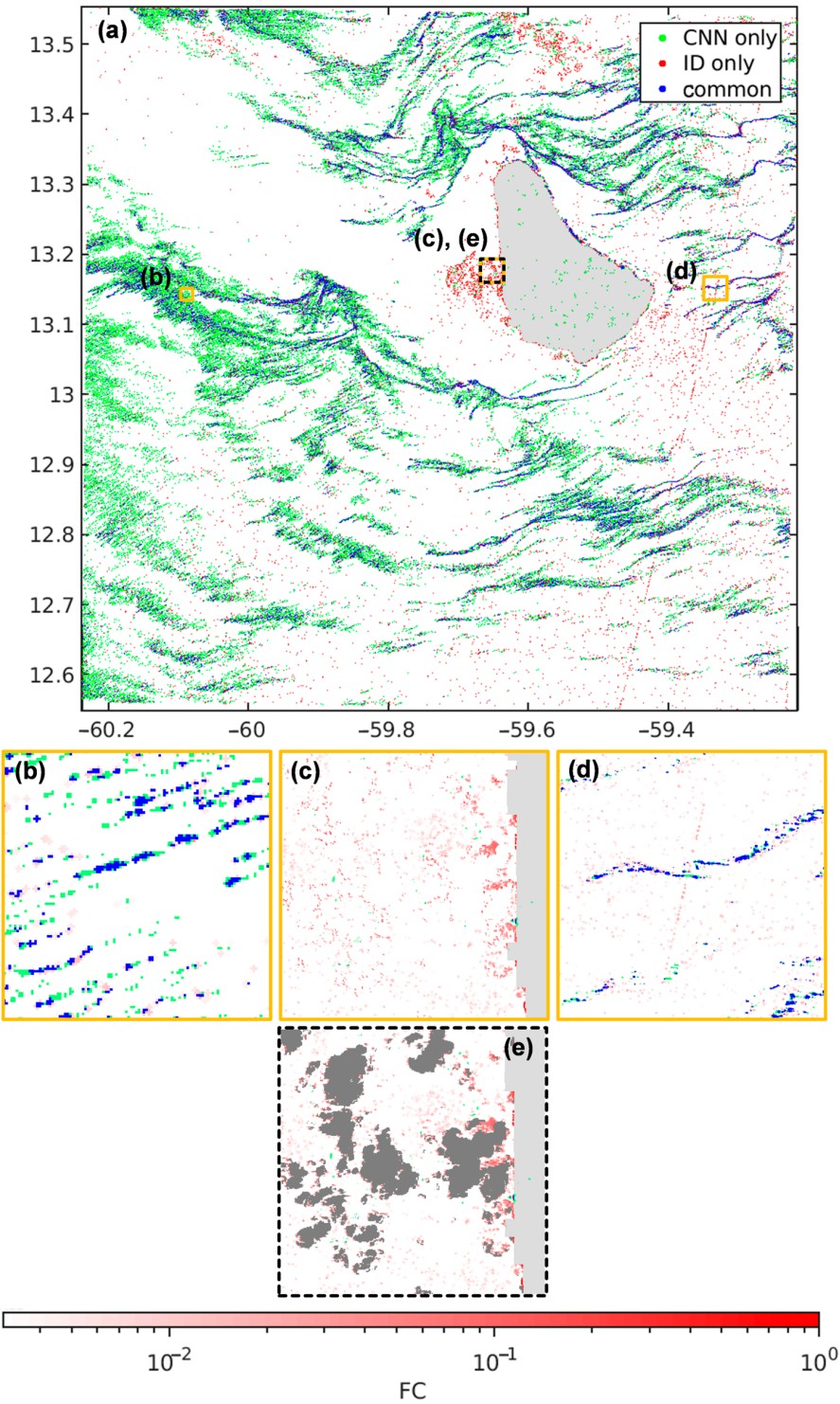

**Figure 6.** (**a**) *Sargassum* detections from MSI with the CNN and ID methods: detected by both methods (blue), detected by the CNN but not by the ID method (green) and detected by the ID method but not the CNN (red). Only FCs above $1.16 \times 10^{-2}$ are represented in (**a**), in order to clarify the figure. Below this threshold the image is too contaminated by a large number of isolated pixels without spatial consistency (discussed in Section 4.2), which start to be visible below this threshold; and (**b**–**e**) sub-images from (**a**) with a color scale for all IDs. FC, cloud (dark gray) and land masks (light gray) come from Descloitres et al. [22]. MSI image, PRV tile from 29 January 2019.

In addition to the main aggregations, the whole FC distribution for all *Sargassum* pixels was similar for the ID and the CNN methods from Sentinel-3 images (ranging from $7.8 \times 10^{-6}$ for low FC to 0.7 for high FC; Figure 7a). A slight and rather constant discrepancy (around 1 to 0.5% of pixels) can be observed.

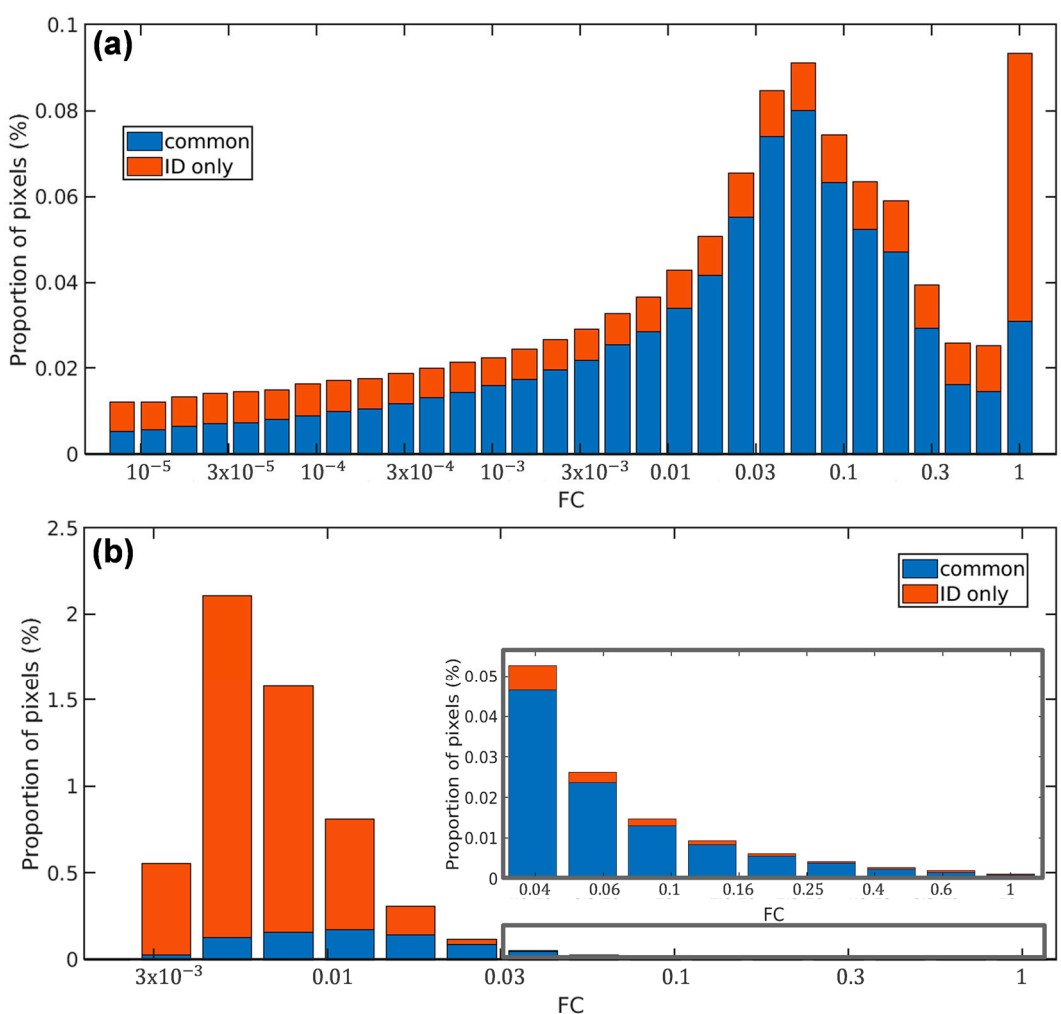

**Figure 7.** Distribution of the proportion of *Sargassum* pixels detected by both the CNN and the ID methods (blue) and detected by the ID method but not the CNN (red) for (**a**) OLCI; and (**b**) MSI.

Regarding MSI, the proportion of common detection depends on the FC (Figure 7b), 35% of the FC detected by the ID method was also detected by CNN. Indeed, only few IDs were detected by the CNN with low FC (Figure 8). Less than 10% of pixels by the ID method were detected as *Sargassum* pixels by the CNN under a *Sargassum* coverage of 0.006 (2.4 m$^2$/pixel). However, this proportion increased with the FC. This detection ability follows a Gompertz curve represented by Equation (9) and shown in Figure 8. About 90% of FC pixels above 0.026 (10.4 m$^2$/pixel) were detected. This curve allows the expected accuracy of *Sargassum* detections to be predicted using the CNN method as a function of FC.

$$Y = c + (d - c) \times \exp(-\exp(b \times (FC - e))) \tag{9}$$

where b = $-170$, c = 4.6, d = 97 and e = $1.25 \times 10^{-2}$.

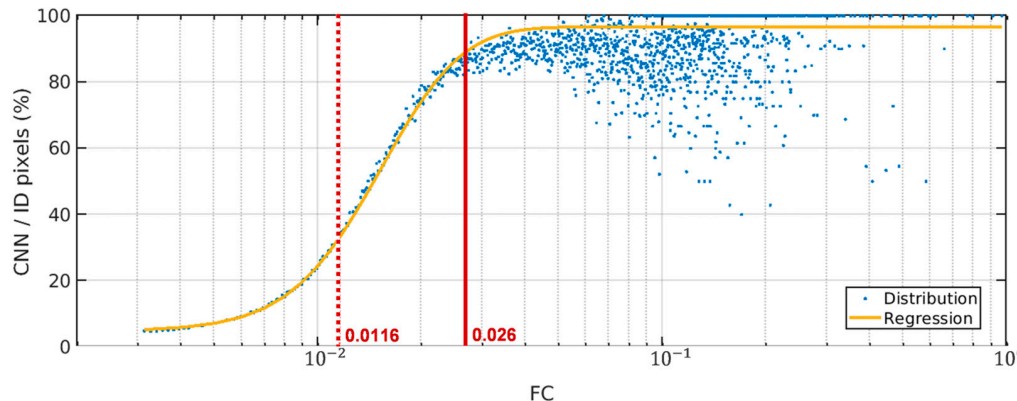

**Figure 8.** Proportion of MSI detections common to CNN and ID methods on the total IDs (in %) as a function of FC intervals (blue dots), each interval has a bin width of $2.79 \times 10^{-4}$. Intervals with less than 10 values are not represented. A Gompertz equation (Equation (9); yellow line) fits the blue points distribution and predicts the CNN detections accuracy. The red plain line represents the FC threshold from at least the CNN detects 90% of the IDs, and the red dotted line is the FC threshold used in Figure 6a.

Nonetheless, a large amount of low FC detections (with an FC under 0.006) with the ID method can either be associated with "true" or "false" detections [21,48] and it may be difficult to estimate the "true" *Sargassum* detection missed by the CNN. Going forward, the CNN can be improved on low index pixels to increase its performance on false positive and true positive pixels of this particular class.

*4.2. Less False Detections Which Improve Sargassum Coverage Estimation*

In the comparison between the two methods, some pixels from the ID method were not detected by the CNN. On MSI images, we only focused on *Sargassum* pixels associated with a FC above 0.026 (see Section 4.1). As presented before, those *Sargassum* pixels seemed to be accurately recognized by the CNN, with more than 90% detections common to the CNN and ID. However, for a few FC classes in that range, the CNN method seemed to underperform compared to the ID. (Figure 8). We focused on these ID extra detections for the MSI, whereas we analyzed all extra detections from the OLCI. Regarding OLCI, on the two OLCI images analyzed, (respectively, for 8 July 2017 and 9 May 2020) about 35% (respectively, 23% and 46%) of the total pixels of the ID method, which correspond to 50% (respectively, 45% and 55%) of the total FC were not detected by the CNN.

As we can easily identify through visual inspection, a large part of those non-detections in OLCI and MSI images should be attributed to false detections from ID methods, which the CNN discards. For instance, haloed false detections around land masses are present on MODIS images using the ID method (Figure 9), and to a lesser extent, on OLCI (Figure 5c) and MSI images (Figure 6c). The ID method also yields very small unrealistic isolated detections, unexpected aggregations in cloudy areas (Figures 5d–g and 6c,e), or artifact patterns linked to the index (satellite swath edges, radiometric noises, isolated detections on satellite image edges) (Figures 5b and 6d). In addition, on OLCI images, the high detection rate for the FC classes closest to one by the ID method (Figure 7a) was poorly recognized by the CNN (only 33% of the class pixels). These extra ID pixels are associated with turbid water that the ID method used here as an undiscarded reference [59]. All the false detection origins presented here are also confirmed by different authors [7,23,24], namely the extra-detection rate in OLCI images is close to the false detection rate observed by Podlejski et al. [24] using the ID method on MODIS images. Such estimations confirm that a large part of the extra-detections is linked to false detection and leads to an overestimation of the coverage of *Sargassum* (FC) by 50%. On MSI images, extra ID detections above a FC of 0.026 are mainly, as for OLCI, false detections obtained by the ID method and

discarded by the CNN. Hence, ID methods for the 0.026 FC threshold seem to induce an overestimation of 14% of the FC for that range. Furthermore, the overestimation can increase by considering the false detections from lower FC pixels (below 0.026).

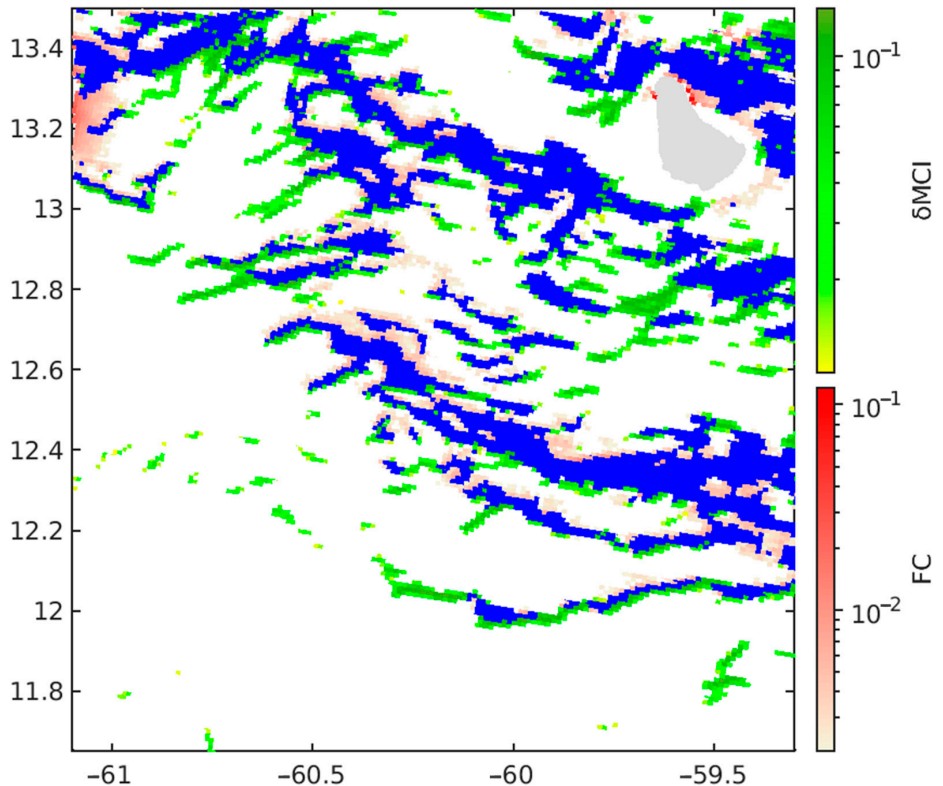

**Figure 9.** Superposition of *Sargassum* detections from MODIS with the ID method and OLCI with the CNN method (δMCI): detected with both MODIS and OLCI (blue), detected with OLCI but not with MODIS (green) and detected with MODIS but not with OLCI (red). Terra and OLCI images from 29 January 2019 at 14:35 UTC and 13:38 UTC respectively.

Therefore, our proposed CNNs discarded false positive detections through an understanding of the spatial context and features (e.g., shape) of *Sargassum* aggregations [24,36,42], leading to a more accurate estimation to the *Sargassum* coverage than the ID method.

### 4.3. Better Estimation of the Aggregation Shape

The CNN generated *Sargassum* pixels not detected by the ID method, which constituted 85% and 56% of *Sargassum* pixels of the CNN dataset for OLCI and MSI, respectively. These extra detections were mainly located in the continuation or along the edges of *Sargassum* aggregations, thus forming bigger aggregations than the ID method. In most cases, these bigger aggregations also included several smaller ID-detected aggregations (Figures 5b–d,f and 6b). The last part of CNN extra detections represented *Sargassum* aggregations not found by the ID method (Figures 5c,d and 6b). The coverage of all CNN extra detections on OLCI were confirmed by the ones found by the ID method on MODIS (Figure 9).

Regarding OLCI images, the CNN method detected four times fewer aggregations than the ID method (around 3000 for the CNN against 11,000 for the ID method). Indeed, the CNN identified larger and longer aggregations that were erroneously detected as several aggregations (Figure 5b–d,f). For instance, on July 8, 2017 (Figure 5a), CNN *Sargassum* aggregations measured, on average 9 km length, versus 2 km for the ID method and covered 50 km$^2$ (CNN) versus 2 km$^2$ (ID). Furthermore, in the case of a blocked signal, such as tiny clouds, the CNN was able to reconnect aggregations with each other and reconstruct

the whole aggregations (Figure 5d,e; lower part). However, this only worked with small clouds (Figure 5d,e; upper part).

Considering MSI images, similarly to OLCI, the CNN detected fewer aggregations than the ID method: ~60,000 versus ~200,000. The CNN aggregations were slightly bigger than the ID-detected aggregations, 5600 m$^2$ and 3600 m$^2$, respectively.

As a result, using our proposed CNNs, the aggregations' shape was more realistic and less discontinuous; more like those observed from satellites of better resolution [49] or from in situ observations [5].

### 4.4. Complementarity of MSI and OLCI Images

*Sargassum* features retrieved by the CNN method using MSI and OLCI differed. The highest-resolution sensor (MSI) had more detailed *Sargassum* aggregations with a shape clearly defined, close to in situ observations [5]. Moreover, it detected *Sargassum* aggregations near the coasts and small aggregations not present in the OLCI images (Figure 10c,d). For instance, in Figure 10a,b, within 1.5 km from the coasts of Barbados Island, around 210 MSI *Sargassum* pixels on the OLCI grid were detected (3200 MSI *Sargassum* pixels), whereas only 80 *Sargassum* pixels are detected using OLCI.

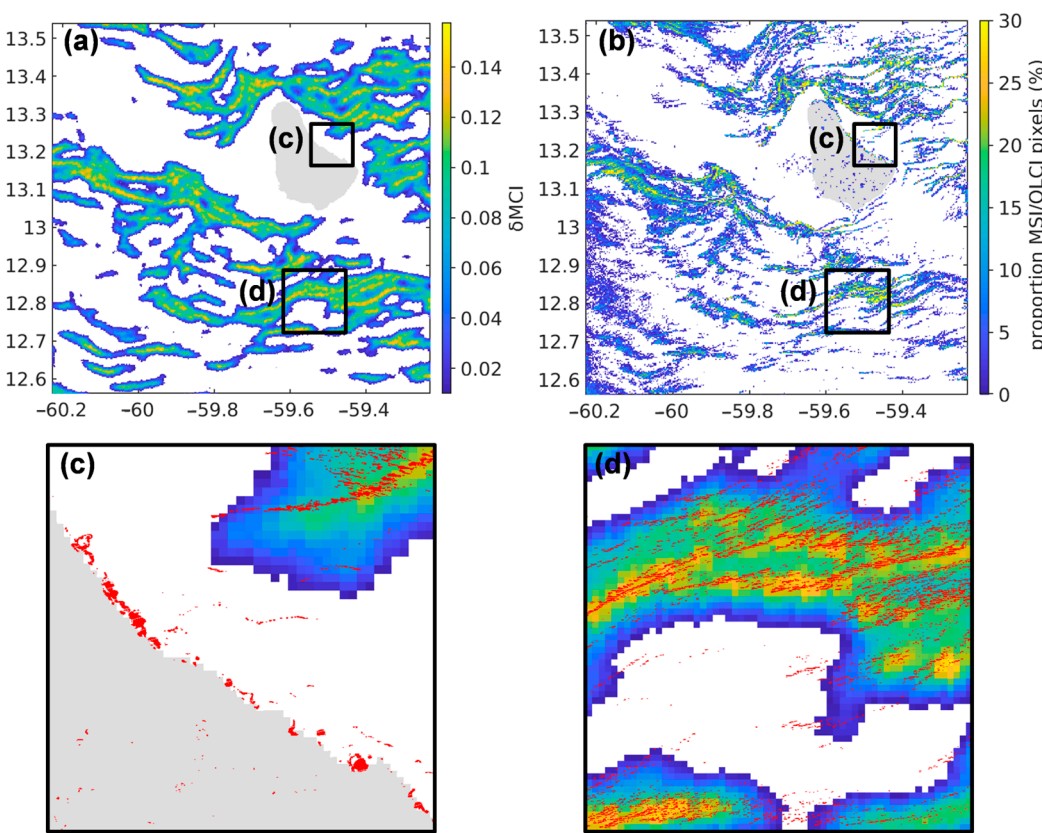

**Figure 10.** (**a**) *Sargassum* detections from CNN with OLCI; (**b**) proportion of MSI *Sargassum* pixels inside 1 OLCI pixel; and (**c,d**) Superposition of *Sargassum* detections from CNN with MSI (red) and with OLCI (color scale); (**c,d**) sub-images from (**a,b**). MSI image, PRV tile and OLCI image from 29 January 2019 at 14:37 and 13:38 respectively.

On the other hand, overall, OLCI has a higher percentage of *Sargassum* pixels than MSI. For instance, in Figure 8, 38% of OLCI pixels are *Sargassum* versus 27% for MSI extrapolated on OLCI pixels (corresponding to 1.89% MSI *Sargassum* pixels). Hence, on average, the CNN detected larger aggregations using OLCI than using MSI (see Section 4.3). Moreover, most MSI detections coincided with higher *Sargassum* signals from the OLCI (Figure S1).

When visually comparing the *Sargassum* aggregations detected from OLCI images with the CNN and MODIS images with the ID method (Figure 9), we retrieved overall the same *Sargassum* aggregations, with a slight shift due to the difference of 1 h between the two sensor acquisition times. The CNN conserved the same proportion of *Sargassum* pixels as MODIS, for instance in Figure 9, 15.7% for the CNN method versus 15.3% for the ID method are *Sargassum* pixels.

OLCI and MSI sensors are complementary. The high resolution MSI sensor detects smaller *Sargassum* aggregations, aggregations nearer to the coasts, and morphology of aggregations with better accuracy. On the other hand, OLCI provides daily data and detects pixels with lower FC.

## 5. Conclusions

The increase in the number of methods and the improvement of the quality of *Sargassum* detection by satellite is crucial for the prediction, and therefore, the management of the future standing of the algae along the coasts. The coupling between satellite detections, ground truth, and modeling remains the best way to understand the dynamics of *Sargassum* along the Great *Sargassum* Atlantic Belt.

In this study, we proposed a new encoder–decoder to detect floating pelagic *Sargassum*. The proposed CNNs were trained using two types of satellite images (OLCI, MSI) with different resolutions. This new model appeared to be more efficient than existing CNNs, such as ErisNet, UNet and SegNet, for *Sargassum* detection, with fewer false positive detections and more accurate *Sargassum* detections. Indeed, the consideration of neighboring pixels avoided some of the false detections made by ErisNet, and fewer reductions improved the performance of UNet.

Our proposed CNNs detected the same large *Sargassum* aggregations detected by the ID method, but with a more accurate estimation of the *Sargassum* coverage. Indeed, with the use of all the spectral bands in the images and the *Sargassum* spatial context, the CNN more efficiently discarded false positive detections as it detected more realistic *Sargassum* aggregations. The *Sargassum* fractional coverage corresponding to the discarded false positive detections was estimated to be 50% for OLCI, and 14% of high FC for MSI.

Furthermore, the CNNs need fewer supplementary post- and pre-treatments than the ID method, and once the model is trained, the use of indexes is not required anymore.

Finally, the study also considered satellite scale characteristics. With MSI, our proposed CNN provided more detailed and distinct aggregations than with OLCI, and was able to detect *Sargassum* aggregations in coastal water with higher confidence thanks to the higher resolution of MSI. The combination of a regional-scale sensor (MSI) and a large-scale sensor (OLCI) may be relevant for the Antilles area, which contains a mix of islands and open sea.

**Supplementary Materials:** The following supporting information can be downloaded at: https://www.mdpi.com/article/10.3390/rs15041104/s1, Table S1: Sentinel-2 MSI images used for the CNN training between 2018 and 2022; Table S2: Sentinel-3 OLCI images of the Lesser Antilles used for the CNN training between 2017 and 2022; Table S3: Detail of our proposed network architecture for MSI images; Table S4: Detail of our proposed network architecture for OLCI images; Figure S1: Distribution of δMCI computed by the CNN.

**Author Contributions:** Conceptualization, M.L., A.B., L.C., J.D., A.S.-G. and C.C.; Data curation, J.D. and C.M.; Formal analysis, M.L., A.B., L.C., J.D. and C.C.; Funding acquisition, R.D. and P.Z.; Methodology, M.L., A.B., L.C., J.D., A.S.-G. and C.C.; Resources, M.L., A.B., L.C., J.D., L.S., A.M.; Software, A.B., L.C. and J.D.; Supervision, C.C.; Validation, M.L., A.B., L.C., J.D. and C.C.; Writing—original draft, M.L., A.B., L.C. and C.C.; Writing—review & editing, A.B., L.C., J.D., L.S., A.M., T.T., R.D., P.Z. and C.C. All authors have read and agreed to the published version of the manuscript.

**Funding:** This research was funded by the National Research Agency (ANR) and by the Territorial Authority of Martinique (CTM) as part of the collaborative FORESEA project (ANR grant number: ANR-19-SARG-0007-07 and CTM grant number: MQ0027405).

**Data Availability Statement:** Not applicable.

**Acknowledgments:** The authors would like to thank Malik Chami for his contribution of the dataset of comparison for OLCI and his relevant suggestions.

**Conflicts of Interest:** The authors declare no conflict of interest.

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
