# Peer review of "Detection of Sargassum from Sentinel Satellite Sensors Using Deep Learning Approach"

_remotesensing, doi:10.3390/rs15041104_

Round 1

Reviewer 1 Report

This manuscript proposes a new deep learning model for Sargassum detection. They used MSI and OLCI images. Their approach outperforms previous deep learning model such as ErisNet, UNet, and SegNet. I think it will be better manuscript if authors revise the following: 

1.    In Abstract, authors should quantitatively reinforce the content of the results.

2.    In Introduction Section, 3 and 4 paragraph parts described the previous researches related to the application of neural network model for Sargassum detection. However, authors did not explain the detailed study area, the used model, and the brief results regarding the previous researches. In addition, authors should add the limitations of the previous researches and the strength of this study. Authors used various deep learning model in this study. Authors need to explain the reason that authors use deep learning model instead of machine learning model.

3.    In 2.1 Study Area part, Figure 1 did not represent latitude and longitude. Authors should add the detailed indicator.

4.    What are the threshold values of each algal index for Sargassum distinction?

5.    2.3.1.2 Section describes the limitation of threshold method for Sargassum detection. 2.3.2.1 Section also explain the trend of state of the art for deep learning model. I think that it would be better to move these parts to Introduction part.

6.    In 2.3.2.2 Section, authors need to add Table for each neural network architecture of MSI and OLCI images. Table should contain block and layer structure/number.

7.    Figure 5 and 6 shows Sargassum detection results from OLCI and MSI with the CNN and ID methods. However, I did not find ground truth Sargassum patches.

8.    In 4.2 Section, authors describe false detection extracted from ID method. What is the specific reason?

9.    In 4.4 Section, it would be good to show the results or advantages of combining MSI and OLCI images. Authors should emphasize the biggest difference between the previous researches and this study.

Reviewer 2 Report

In this work, two Convolutional Neural Networks (CNN) are proposed with an encoder-decoder architecture, for Sargassum detections on coastal and offshore waters based on all the spectral bands provided by MSI on Sentinel 2 and OLCI on Sentinel 3, which have different number of bands and different spatial resolution. The Study area is located in the Lesser Antilles, in the East Caribbean Sea and the Mexican coast.

Performance of a new algorithm is compared with performance of indices: NDVI, Maximum Chlorophyll Index (MCI), Floating Algae Index (FAI), Alternative Floating Algae Index (AFAI) and two previously developed NN approaches, showing better performance of the proposed method in many cases. Limitations of the method are also discussed.

The manuscript is well written and can be published without changes.

Author Response

We would like to thank the reviewer for his comments

Round 2

Reviewer 1 Report

Dear authors,

I confirmed the answers regarding my comments.

I think that the answers are appropriate well. However, I hope that the authors revise Figure 1 to general type (ex. 24oN or 119oE).

That's all.

Author Response

We would like to thank the reviewer for his feedback.
As suggested, we have replaced the coordinates in Figure 1